# Effects of Different Land Use Types and Soil Depths on Soil Mineral Elements, Soil Enzyme Activity, and Fungal Community in Karst Area of Southwest China

**DOI:** 10.3390/ijerph19053120

**Published:** 2022-03-07

**Authors:** Jiyi Gong, Wenpeng Hou, Jie Liu, Kamran Malik, Xin Kong, Li Wang, Xianlei Chen, Ming Tang, Ruiqing Zhu, Chen Cheng, Yinglong Liu, Jianfeng Wang, Yin Yi

**Affiliations:** 1Key Laboratory of National Forestry and Grassland Administration on Biodiversity Conservation in Karst Mountainous Areas of Southwestern China, Guizhou Normal University, Guiyang 550025, China; 201307048@gznu.edu.cn (J.G.); lanzhoudaxue2022@163.com (J.L.); kongxin1232022@163.com (X.K.); gznu_wangli0521@163.com (L.W.); chenxianlei0321@163.com (X.C.); mingtang@gznu.edu.cn (M.T.); 2State Key Laboratory of Grassland Agro-Ecosystems, Center for Grassland Microbiome, Lanzhou University, Lanzhou 730000, China; houwp19@lzu.edu.cn (W.H.); chengch20@lzu.edu.cn (C.C.); liuyl2020@lzu.edu.cn (Y.L.); 3College of Pastoral Agriculture Science and Technology, Lanzhou University, Lanzhou 730000, China; malik@lzu.edu.cn; 4Qinghai Provincial Key Laboratory of Medicinal Plant and Animal Resources of Qinghai-Tibet Plateau, Academy of Plateau Science and Sustainability, School of Life Sciences, Qinghai Normal University, Xining 810008, China; zrq15002679150@163.com; 5Collaborative Innovation Center for Western Ecological Safety, Lanzhou University, Lanzhou 730000, China; 6State Key Laboratory of Plateau Ecology and Agriculture, Qinghai University, Xining 810016, China

**Keywords:** karst areas, soil depth, land use types, soil metal elements, soil enzyme activity, soil fungal community and diversity

## Abstract

The current research was aimed to study the effects of different land use types (LUT) and soil depth (SD) on soil enzyme activity, metal content, and soil fungi in the karst area. Soil samples with depths of 0–20 cm and 20–40 cm were collected from different land types, including grassland, forest, *Zanthoxylum planispinum* land, *Hylocereus* spp. land and *Zea mays* land. The metal content and enzyme activity of the samples were determined, and the soil fungi were sequenced. The results showed that LUT had a significant effect on the contents of soil K, Mg, Fe, Cu and Cr; LUT and SD significantly affected the activities of invertase, urease, alkaline phosphatase and catalase. In addition, Shannon and Chao1 index of soil fungal community was affected by different land use types and soil depths. Ascomycota, Basidiomycota and Mortierellomycota were the dominant phyla at 0–20 cm and 20–40 cm soil depths in five different land types. Land use led to significant changes in soil fungal structure, while soil depth had no significant effect on soil fungal structure, probably because the small-scale environmental changes in karst areas were not the dominant factor in changing the structure of fungal communities. Additionally, metal element content and enzyme activity were related to different soil fungal communities. In conclusion, soil mineral elements content, enzyme activity, and soil fungal community in the karst area were strongly affected by land use types and soil depths. This study provides a theoretical basis for rational land use and ecological restoration in karst areas.

## 1. Introduction

Karst landform is a variety of peculiar landforms formed on the surface and underground under the continuous dissolution of a large number of soluble rocks by flowing water. Karst area accounts for about 12% of the world’s land area [1]. Due to the unique geographical conditions, the southwest karst region centered on Guizhou Province in China has the largest continuous coverage area of about a 5.1 million km^2^ in the world. It is also the most complete development type and typical karst ecosystem in the world [2]. However, the karst ecosystem is highly heterogeneous and fragile, and can cause rocky desertification [3]. Here, arable land is limited by rocks, and soil functions and ecosystem services are negatively affected by poorly managed land use patterns [4].

Land use, as a comprehensive reflection of human behaviors, is closely related to plant communities, soil nutrients and soil enzyme activities, resulting in differences in soil microbial characteristics [5]. Studies have confirmed that different land use patterns showed significant effects on soil nutrients in karst areas of Southwest China [6]. There were differences in community structure/diversity of soil fungi in different vegetation succession stages in the karst area and the soil bacterial community had complex responses to tillage patterns [7]. Both soil enzyme activity and microbial community structure depend on land use patterns. Therefore, in this case, land use conversion in the karst area has become a major issue. Additionally, different depths of the soil layer influence soil microorganisms, vegetation types, and litter quality, which leads to the difference in microbial community structure. Few studies claimed that microbial respiration activity, biomass and diversity were decreased with soil depth and the number of bacteria in the topsoil layers of three different land types was 2–20 times more than the lower layer in the karst area compared to the non-karst [8,9]. Hence, the change of soil nutrients from top to bottom in the karst area is very clear. In recent years, local farmers have been encouraged to grow different cash crops, to improve the ecological environment of the karst area. Therefore, exploring the effects of different land use patterns and soil layers is helpful for ecological restoration in this fragile ecological area.

Soil enzymes have a high catalytic capacity and take part in organic matter decomposition and cycling of nutrient in ecosystem [10]. Soil nutrients, microorganisms, vegetation types and management measures affect soil enzyme activities in different degrees [11,12,13]. More importantly, soil enzyme activity can represent the rate of nutrient uptake by microorganisms and plants. Enzymes are sensitive to reflect the early changes of soil quality caused by soil management [14]. Soil invertase plays a vital role in C decomposition, transformation, and soil bio-respiration [15]. Urease converts organic N to available N by hydrolyzing urea [16]. In addition, alkaline phosphatase plays an important role in organic phosphorus (P) mineralization and plant P nutrition, especially in calcareous soil with limited P [17]. Microbial activity is a key indicator to detect soil quality and to control land degradation which fully shows that soil microorganism is still an important research direction, especially in karst areas. However, the majority of the studies focused on the soil bacterial community in the karst area, such as the response of soil bacterial community structure to different disturbances [18] and the correlation between vegetation succession and bacterial metabolic diversity [19]. Fungi are the basic components of the soil microbial community, and it is necessary to investigate the relationship between soil fungal community and land use types.

Heavy metals are widespread on the earth’s surface which are persistent, stable, and difficult to degrade [20]. In recent years, due to unreasonable exploitation of mineral resources, improper disposal of hazardous wastes, and the extreme vulnerability of groundwater systems, heavy metals in karst areas have been seriously diffused, posing a serious threat to the biological community [21,22]. Studies have shown that land use can directly or indirectly affect the content of heavy metals by changing soil properties [23,24]. Soil enzymes produced by microbial metabolism can be used as monitoring factors for heavy metals. Soil mineral elements are also important indicators to determine soil fertility and the healthy growth of plants. For instance, potassium and sodium are closely related to the soil microbial community in the karst ecosystem [25].

Land use conversion is a part of China’s policy of “Grain for Green Project”; however, little is known about the effects of different land use patterns and soil depths on mineral elements content, enzyme activities and fungal communities in karst areas, Southwest China. In the current study, the contents of soil mineral elements, soil enzyme activities and fungal communities in five land use types and two soil depths in Guizhou Province of China were investigated to provide a theoretical basis for soil management and ecological restoration in karst areas.

## 2. Materials and Methods

### 2.1. Study Area and Soil Sampling

Huajiang town is located in Guanling Buyi and Miao Autonomous county, Anshun city, Guizhou province, southwest China with an area of 294.9 km^2^ (Altitude 1439 m, 105°34′ E, 25°43′ N), which is a typical karst landform. The study site is dominated by the humid subtropical monsoon climate, and the annual mean temperature and rainfall are 17 °C and 1200 mm per year, respectively, while the frost-free period is about 288 days. In November 2019, we selected five land use types, including: grassland (the main species are *Themeda japonica* and is located in 105°28′42″, 25°43′48″, altitude 851 m), secondary forest (the main species are *Liquidambar formosana* and is located in 105°28′46″, 25°43′48″, altitude 798 m); pepper field (the main cultured species are *Zanthoxylum planispinum* and is located in 105°39′58″, 25°40′07″, altitude 517 m), dragon fruit field (the main cultured species is *Hylocereus* spp. and locate in 105°39′41″, 25°40′33″, altitude 598 m) and maize field (the main cultured species are *Zea mays* and is located in 105°39′42″, 25°40′33″, altitude 798 m). There were 5 sampling points for each land use type, the area of each sampling point is 4 m × 4 m, and the distance between each sampling point was 5 m. After removing 1–2 cm of topsoil, soil samples were collected at depths of 0–20 cm and 20–40 cm. Each soil sample was mixed from four subplots (1 m × 1 m range) at 0–20 cm and 20–40 cm soil depths, respectively, and collected in dry, clean, sterile polyethylene bags. Next, all soil samples were passed through a 2 mm sieve to remove visible roots and stones. A portion of each soil sample was transferred in the 50 mL sterile centrifuge tube and placed in a liquid nitrogen tank. The samples were transported to the laboratory, and the sterile centrifuge tube was immediately stored in an ultra-low temperature refrigerator at −80 °C for subsequent soil enzyme activity measurement and soil DNA extraction. The other portion of the soil samples were kept for natural drying to determine the content of soil mineral elements.

### 2.2. Soil Mineral Elements Content Assay

The contents of soil mineral elements include potassium (K^+^), calcium (Ca^2+^), sodium (Na^+^), magnesium (Mg^2+^), iron (Fe^3+^), copper (Cu^2+^), zinc (Zn^2+^), cadmium (Cd^2+^), chromium (Cr^2+^) and lead (Pb^2+^) were determined by atomic absorption spectroscopy method after digestion according to the method described by Li et al. [26] The contents of soil mineral elements were measured by TRACE AI1200 atomic absorption spectrometer (Canada Aurora, Vancouver, BC, Canada).

### 2.3. DNA Extraction and Illumina Sequencing

Soil DNA (300 mg) was extracted using the power soil DNA extraction and separation kit according to the protocol of the manufacturer (MoBio, Carlsbad, CA, USA). The ITS1 region of the fungal rRNA gene was amplified by PCR as described by Zhong et al. [27]. The primers were ITS1 (5′-CTTGGTCATTTAGAGGAAGTAA-3′) and ITS2 (5′-GCTGCGTTCTTCATCGATGC-3′). The PCR procedure consisted of 27 cycles at 94 °C for 2 min, 94 °C for 30 s, 55 °C for 30 s, and 72 °C for 60 s, and 72 °C for 10 min. PCR products were sent to Genesky Biotechnologies Inc., Shanghai, 201,315 (China) for sequencing with an Illumina 2 × 250 bp platform.

### 2.4. Sequencing Data Processing and Analysis

The original sequence was filtered using the method described by Caporaso et al. [28] to eliminate the low-quality sequence, and then the ITS2 region was extracted by the same method used by Bengtsson-Palme et al. [29]. In the subsequent examination of potential chimeras, the uchime command in mothur version 1.31.2 [30] was used and compared with entries in the DNA based fungal species unified system related to the classification (unite) database [31]. Finally, after dereplicating and discarding all monomers, the non-chimeric sequences were aggregated into the operational taxonomic units (OTUs), and the OTUs were clustered based on the UPARSE pipeline using USEARCH version 8.0 (the similarity of OTU is 97% [32].

### 2.5. Soil Enzymes Assay

The activities of soil invertase (Inv), urease (Ure), and alkaline phosphatase (Alp)was measured by 3,5-Dinitrosalicylic acid colorimetry, sodium phenol-sodium hypochlorite and disodium diphenyl phosphate colorimetry, respectively according to description of Hou et al. [33]. The activity of soil catalase (Cat) was determined by Potassium permanganate titration according to Chao et al. [34].

### 2.6. Statistical Analyses

The data were analyzed using SPSS software 17.0 (IBM Inc., Armonk, NY, USA). The effects of land use type (LUT) and soil depth (SD) on the soil fungal alpha diversity (Chao1 and Shannon indices), soil mineral elements content and soil enzyme activity were analyzed by two-way ANOVA. Statistical significance was defined as *p* = 0.05 confidence level, and the mean was evaluated by the standard error. The Chao1, Shannon, the heatmap, PCoA, redundancy analysis (RDA), Variance Partitioning Analysis (VPA) and Spearman’s rank correlation analysis were carried out in R (version 3.2.2). The Linear discriminant analysis (LDA) coupled with effect size measurement (LEfSe) analysis was performed using the OmicStudio tools at https://www.omicstudio.cn/tool (accessed on 27 December 2021).

## 3. Results

### 3.1. Differences in Content of Soil K, Na, Ca, Mg and Fe in Different Land Use Types and Soil Depths

Appendix A presented data on soil K, Na, Ca, Mg and Fe contents as influenced by land use types (LUT) and soil depth (SD). Our results showed that LUT have significant effects on soil K (*p* < 0.001), Na (*p* = 0.029), Ca (*p* = 0.01), Mg (*p* < 0.001) and Fe (*p* < 0.001) contents. However, SD had no remarkable role in the content of all soil mineral elements. Additionally, the LUT × SD interaction only had a clear effect on Ca (*p* < 0.001) and Mg contents (*p* < 0.001) (Appendix A). Further, we found that there was no significant difference in the K, Na, and Mg content between 0–20 cm and 20–40 cm soil layers under the five different land use types (Figure 1b,c,e). The soil Ca content at 0–20 cm depths in grassland and *Zea mays* was 2.3-fold higher and 1.4-fold lower than that at 20–40 cm depth, respectively (Figure 1d). Interestingly, the Fe content was significantly 1.6 times higher only in the 0–20 cm *Zea mays* soil compared to the 20–40 cm soil layer (Figure 1f). Different land use types also have different effects on soil mineral elements content. Significant differences in soil K, Mg, and Fe contents were found between land use types at both soil depths; however, land use type played no clear role on Na content in different soil layers (Figure 1). The highest levels of K were found in *Zanthoxylum planispinum* soils, while Mg and Fe contents were highest in grassland soils at both soil depths. Meanwhile the contents of K in grassland soil and Fe in *Zanthoxylum planispinum* soil were the lowest, (Figure 1b,e,f). Whereas, at the deep soil layer (20–40 cm), the Ca content was highest in the grassland soil and lowest in the *Zea mays* soil (Figure 1d).

### 3.2. Differences in Content of soil Cr, Ni, Cu, Zn, Cd and Pb in Different Land Use Types and Soil Depths

As shown in Appendix A, soil Cr (*p* = 0.033), Cu (*p* < 0.001), Zn (*p* = 0.011) and Cd (*p* < 0.001) contents were obviously influenced by LUT. However, soil heavy metal levels were not significant to SD. Interestingly, LUT × SD interaction only affected soil Cu content (*p* = 0.026) (Appendix A). Subsequent studies have shown that in the five different land use types, there was no significant difference in heavy metal content between different layers (Figure 2). However, we found that all heavy metal contents in deep soil layers were significantly affected by land use types. The highest content of Cr, Ni, and Pb existed in secondary forest soil and the lowest existed in *Zea mays* soil (Figure 2a,b,f). The contents of Cu, Zn and Cd in grassland soil were significantly lower than the other land use types (Figure 2c–e). Moreover, soil Cu and Cd contents in *Zanthoxylum planispinum* and Zn content in the forest were the highest (Figure 2c–e). It should be noted that in the topsoil (0–20 cm), different land use types only affected the contents of Cu and Cd, and the contents of Cu and Cd in *Hylocereus* spp. soil were significantly higher than those in other land types (Figure 2c,e).

### 3.3. Differences in Soil Enzyme Activity in Different Land Use Types and Soil Depths

Statistical evaluation of the effects of land use types on enzyme activity were statistically significant (*p* < 0.05) as shown in Appendix A, which the role of soil depth was considered. LUT significantly affected the activity of Inv (*p* < 0.001), Ure (*p* < 0.001), Alp (*p* < 0.001). SD and LUT × SD interaction all had extremely significant influence on the activity of the four enzymes (*p* < 0.001) (Appendix A). With the increase in soil depth, the enzyme activities of different land use types all showed a downward trend. In grass land, *Zanthoxylum planispinum* and *Hylocereus* spp. soil, Inv activity in 0–20 cm soil decreased by 1.5 times, increased by 1.7 times and 3.6 times compared with that in 20–40 cm soil depths, respectively (Figure 3a). The activity of Ure at soil depths of 0–20 cm in grassland, forest, and *Zanthoxylum planispinum* was 1.7, 4.3 and 12.9 times higher than that in 20–40 cm soil, respectively, but it was 5.7 times lower in 0–20 cm soil compared to 20–40 cm depths in *Zea mays* soil (Figure 3b). The Alp activity in 0–20 cm depth was increased by 2.2, 1.7 and 1.6 times compared to 20–40 cm depth in forest land, *Hylocereus* spp. and *Zea* mays soil, respectively (Figure 3c). Similarly, Cat activity in topsoil of *Hylocereus* spp. and *Zea mays* soil was significantly higher than that in deep soil layers, which was 1.9 and 1.8 times, respectively (Figure 3d). Further, we studied the response of soil enzyme activities to land use types at different depths. At the two soil depths, different land use types had significant effects on soil enzyme activity. The activity of Inv and Ure was highest at 0–20 cm depths of *Zanthoxylum planispinum* lowest in grass land. In 20–40 cm soil layer, Inv activity was the highest in forest, and the lowest was found in *Hylocereus* spp. soil(Figure 3a). Urease activity was the lowest in 0–20 cm layer of *Zea* mays soil. On the contrary, in 20–40 cm layer, its activity was the highest in *Zea* mays soil and the lowest in grass land (Figure 3b). In shallow soil, the activity of Alp was the highest in forest, and the lowest in *Hylocereus* spp.. Interestingly, the lowest activity of Alp and catalase in deep soil all existed in *Hylocereus* spp. soil (Figure 3c,d).

### 3.4. The Richness and Diversity of Soil Fungal Community

We found that LUT and SD had an obvious effect on the Shannon index (*p* = 0.018; *p* = 0.004, respectively, Appendix A), but the Shannon index was not significantly affected by the interaction of LUT and SD. Similarly, LUT had a marked influence on the Chao 1 index (*p* = 0.028, *p* = 0.03, respectively, Appendix A), and LUT × SD caused a significant influence on the Chao 1 (*p* = 0.032, Appendix A). Our results also showed that Shannon and Chao 1 index were affected by different soil depth and land use types. In *Zea* mays, the Shannon and Chao 1 index in the topsoil layers are all significantly higher than that of the deep soil layers, and the Shannon and Chao 1 of topsoil layers in *Zea* mays land were 1.21 times and 1.42 times higher than those of deep soil layers, respectively (Figure 4a,b). However, the rest of the land use types did not change significantly with the change of soil depth. Additionally, Shannon at a soil depth of 0–20 cm in *Zanthoxylum planispinum* land was the highest (4.87 ± 0.15) and was the lowest in *Hylocereus* spp. land (3.86 ± 0.15) (Figure 4a). The same result was shown in the Chao 1. Chao 1 at 0–20 cm depth in *Zanthoxylum planispinum* land was the highest (579.20 ± 22.39) and *Hylocereus* spp. land was the lowest (385.00 ± 29.85) (Figure 4b).

### 3.5. Relative Abundance of Soil Fungal Community

Figure 5a,b showed the relative abundances of major fungal phyla and genera in 0–20 cm and 20–40 cm soil depth in five different land use types, respectively. The Ascomycota, Basidiomycota, and Mortierellomycota were all the dominant phyla at 0–20 cm and 20–40 cm soil depths in the five different land use types (Figure 5a). Interestingly, in 20–40 cm depth soil of grassland, the relative abundance of Basidiomycota was the highest (Figure 5a). From the level of genus, the *Fusarium*, *Mortierella* and *Tetracladium* were the dominant genus in 0–20 cm and 20–40 cm soil depths in forest and grassland (Figure 5b). *Fusarium* and *Mortierella* were the dominant genus in 0–20 cm and 20–40 cm soil depths in *Hylocereus* spp., *Zanthoxylum planispinum* and *Zea* mays (Figure 5b). Furthermore, *Preussia* was also the dominant genus at two soil depths of *Zea* mays (Figure 5b).

At the genus level, the heat map produced by R also showed the differences of soil fungal community aggregation patterns at 0–20 cm and 20–40 cm depths in different land use types (Figure 6). Compared with the other four land use types, the vast majority of the top 30 soil fungal genera in 0–20 cm and 20–40 cm depths soil in *Zea* mays land have higher absolute abundance (Figure 6). In contrast, the absolute abundance of most fungal genera in forest and grassland at 0–20 cm and 20–40 cm soil depths was significantly lower than that of the other three different land use types (Figure 6). Besides, we determined the beta diversity to study the effects of different land use types and soil depth on soil fungal communities (Figure 7). The first principal component explained 21.83% of the total variance, and the second principal component explained 10.79% of the variance. The points of five different land types were significantly dispersed, indicating that land use type caused the significant change of soil fungal community structure. However, except for *Zea mays*, the points of 0–20 cm depths soil and 20–40 cm depths soil were very concentrated, indicating the similarity of fungal community composition at the two soil depths, and the effect of soil depth on soil fungal structure was not significant.

### 3.6. Comparison of Different Fungal Species at 0–20 cm and 20–40 cm Soil Depth in Different Land Use Types

Histogram of LDA value distribution of soil fungi at different depths in different land use types. Appendix A showed the significantly different species with LDA score greater than the preset value (the default preset value was 3.0). In the 20–40 cm depth soil of grassland, the abundance of fungi of only one genus was significantly higher than that of 0–20 cm depth which was Ochroconis (Appendix A). In the topsoil layers of *Zanthoxylum planispinum*, the relative abundance of Glomeromycota and Glomus was significantly higher than that at the soil depths of 20–40 cm (Appendix A). In *Hylocereus* spp. land, the relative abundance of Tecladium in 0–20 cm soil was significantly higher than that in 20–40 cm soil (Appendix A). In *Zea* mays soil, the relative abundance of Humicola in 0–20 cm soil layer was significantly higher than that in 20–40 cm soil layer, while the relative abundance of Gibberella in 20–40 cm soil layer was significantly higher than that in 0–20 cm soil layer (Appendix A). We also found that there were no fungal groups with significant differences in relative abundance in different soil depths of forest.

### 3.7. Correlation of Soil Fungal Communities Composition and Diversity with Content of Soil Mineral Elements

At soil depths of 0–20 cm and 20–40 cm, there was significant relevance between soil fungal diversity and soil mineral contents of different land use types (Appendix A). We found that at depth of 0–20 cm soil, the contents of Cr, Ni, Cu (*p* < 0.05) and Zn, Cd and Pb (*p* < 0.001) in grassland were significantly correlated with Shannon index (Appendix A); the content of Mg (*p* < 0.05) in forest was significantly related to Chao1 index (Appendix A). Similarly, the contents of soil Ni, Cu, Cd and Pb (*p* < 0.05) were closely related to PCoA in *Zanthoxylum planispinum* (Appendix A), and in *Hylocereus* spp., PCoA was closely related to Mg content (*p* < 0.05 Figure 8g). At soil depths of 20–40 cm, PCoA was closely related to K content in forest (*p* < 0.05 Appendix A); in *Hylocereus* spp. soil, Chao1 was significantly correlated with K content, but PCoA was significantly correlated with contents of Cr (*p* < 0.05), Ni (*p* < 0.01), Zn (*p* < 0.05), Cd (*p* < 0.05) and Pb (*p* < 0.01 Appendix A). We also found that Shannon and Chao 1 index was clearly related to K content (*p* < 0.01) and Ni content (*p* < 0.05) in *Hylocereus* spp. soil (Appendix A). Morever, in *Zea* mayssoil, PCoA was significantly correlated with contents of Cd and Pb (*p* < 0.05 Appendix A).

At soil depths of 0–20 cm, there was a significant positive correlation between K content and Blastocladiomycota; Na content also had a positive effect on Blastocladiomycota and negative effect on Chytridiomycota; Ca content had a positive impact on Zoopagomycota; Fe content had a positive effect on Glomeromycota and negative effect on Basidiomycota; Content of Cd and Pb had a positive effect on Mortierellomycota and Rozellomycota; interestingly, the content of Cr, Ni, Cu and Zn had no significant effect on the types of fungi (Figure 8a,c,e,g,i). RDA also showed that in topsoil layers, contents of Ca, Fe and Cu were the main factors affecting the change of soil fungal community in *Hylocereus* spp. soil; K was an important factor to affect the changes of soil fungal community in *Zanthoxylum planispinum* soil; Cd and Na were primary factors to affect soil fungal community in *Zea mays* soil, and Mg and Ca were principal considerations affecting the changes of fungal community in grassland and *Hylocereus* spp. soil (Figure 9a). At 20–40 cm soil depths, K content had a strong negative effect on Ascomycota and a positive effect on Calcarisporiellomycota; Ca content had a negative effect on Blastocladiomycota and Basidiomycota; the content of other heavy metals had a strong negative effect on Basidiomycota and Rozellomycota (Figure 8b,d,f,h,j). In addition, in deeper soil, K was the important factor to affect the changes of soil fungal community in *Zea mays* soil; content of Ca, Fe, and Mg were the main factors affecting the change of soil fungal community in grassland and forest; and heavy metal content were important factors to affect soil fungal community in *Zanthoxylum planispinum* soil (Figure 9b). 

### 3.8. Correlation of Soil Fungal Communities Composition and Diversity with Soil Enzyme Activity

Similarly, our result showed that under the two soil depths, there was significant relevance between soil fungal diversity and soil enzyme activity of different land use types (Appendix A). At 0–20 cm soil depths, Shannon index was related to Inv activity in grassland (*p* < 0.05 Appendix A) and Cat activity in *Zea* mays soil (*p* < 0.05 Appendix A), and in *Zea* mays soil, Chao1 was also closely related to Cat activity (*p* < 0.01 Appendix A). At soil depths of 20–40 cm, the activity of Cat was apparently correlated with Shannon index and PCoA (*p* < 0.05 Appendix A); in *Zanthoxylum planispinum*, Shannon index was markedly correlated with Alp activity (*p* < 0.01 Appendix A), and in *Hylocereus* spp. soil, Chao1 was significantly correlated with Inv activity (*p* < 0.05 Appendix A).

In different soil depths, the response of fungal phylum level categories of different land use types to the change of soil enzyme activity was also different (Figure 10). In topsoil layers, we found that there was a clear positive correlation between Ure activity and Calcarisporiellomycota and Chytridiomycota; Cat activity had a strong positive effect on Mortierellomycota; interestingly, in forest Ure activity had a positive effect on Ascomycota and Chytridiomycota, but in *Hylocereus* spp. soil, Ure activity had a strong negative effect on Ascomycota; and Inv activity had no distinct effect on all the phyla levels (Figure 10a,c,e,g,i). In deep soil layers, Inv activity had a strong positive effect on Chytridiomycota; Ure activity had a positive effect on Calcarisporiellomycota and a negative effect on Ascomycota; Alp activity had a positive effect on Ascomycota and a passive effect on Chytridiomycota, Glomeromycota and Kickxellocymota; and Cat activity had a strong effect on Basidiomycota (Figure 10b,d,f,h,j). RDA analysis also showed that in 0–20 cm soil depth layer, the activities of Inv and Alp were the main factors affecting soil fungal communities in forest, *Zanthoxylum planispinum* and *Zea mays* soil; Cat activity was an important factor affecting soil fungal community in *Hylocereus* spp. soil, such as Zoopagomycota and Ascomycota; Ure mainly affected grassland soil Blastocladiomycota (Figure 11a). At 20–40 cm soil depths, the activities of Inv and Alp were the main factors affecting the fungal community in forest and *Zanthoxylum planispinum* soil; Cat was the main factor affecting soil fungal communities in grassland, forest, and *Zanthoxylum planispinum* land; Ure had a strong positive effect on *Zea mays* land soil, such as Rozelomycota, Mortierellomycota, and Kickxelomycota (Figure 11b).

The composition and diversity of soil fungal community in different types of land use and soil depth changed, and the change depended on soil mineral elements contents and soil enzyme activity. However, the contribution of these factors to the soil fungal community is still unclear. Therefore, the bonding contributions of soil mineral elements and soil enzyme activities in the fungal communities were also investigated by VPA analysis. The result revealed that land use types, soil depth, soil mineral elements content, and soil enzyme activity explained 12%, 0.9%, 1.2%, and 0.4% on the total variations of the fungal community, respectively (Figure 12).

## 4. Discussion

Different land use types can lead to different ecosystem functions by influencing underground processes, and soil mineral elements can be affected by soil properties and land management and utilization [35,36]. It has been reported that grassland reclamation in the karst area reduced soil trace elements (Cu, Fe, Mo, B) and enriched in surface soil [37]. Li et al., found that the land use history changed the central trend and heterogeneity of soil properties, including C, N, C:N, Ca, and K [38]. Similarly, land use patterns affect plant litter, thus changing the microbial activities, leading to changes in soil nutrients, and informing a typical correlation among soil microorganisms, vegetation, soil nutrient, and mineral [39]. In the current study, in two different depths of soil K content of different land use types from high to low was followed *Zanthoxylum planispinum* land > *Hylocereus* spp. land > *Zea mays* land > forest land > grassland at two different depths of soil. Similar studies also observed that the K content in cultivated soil was higher than the other types of land [40]. *Zanthoxylum planispinum* and *Hylocereus* spp. are economic crops with a large amount of fertilizer, which may accelerate nutrient turnover, promote K accumulation, or increase availability [41]. Yu et al., also confirmed that chemical fertilizers can effectively increase the total K and P content in soil [42]. The natural karst environment is rich in Ca, which has a remarkable effect on the soil physical and chemical properties [43]. We found that the Ca content was the highest in grass land at soil depths of 20–40 cm, followed by secondary forest. Studies showed that the organic matter and humic acid of karst soil have strong adsorption and complexation to Ca [44,45]. A large amount of litter in grassland and secondary forest returned to the soil and provided a C source. The input of litter is related to the richness of the forest. The plant tissue falling from tulip poplar constituted a large amount of organic matter input into the soil, which contributes to the low variability of C and N concentration on a small scale, while the Ca accumulated in plant tissues enables the trees to absorb it from deep soil layers to maintain the effective calcium concentration in topsoil layers, which eventually cause the reduction in local soil Ca variability [46]. Similarly, the contents of Mg and Fe was highest in the grassland at two different soil depths, while they were reduced in different tillage systems and planting patterns. The possible reason for the decline may include: (1) the coarsening of soil particle composition resulted in the decrease in element adsorption, resulting in the leaching of mineral nutrients, (2) the decomposition of organic matter accelerated, and pH changed, which affected the deposition of elements, (3) Fe was mainly controlled by soil formation and weathering, and the degree of weathering was high in the farming area [41,47]. Our results showed that the five different land management types had no significant effect on Na content, similar to Yu et al. [41]. The differences in field management methods, such as fertilizer application quantity and mode can lead to variations in soil physical and chemical properties and nutrient availability under different tillage systems and planting modes [48]. Interestingly, we found that soil depth had no significant effect on mineral nutrient contents. However, the content of Ca and Fe in 0–20 cm soil layer was significantly higher than that in the 20–40 cm soil layer probably because the soil biological activities were frequent in the thick maize roots that contributed to deep absorption. Ca content in the deep soil layer was higher than the shallow layer of grassland, which may be related to the return of grass root biomass and the decrease in surface runoff [49].

Land use types correspond to the changes in soil management, vegetation types and microbial activity. These changes have an important impact on the migration, transformation, and enrichment of soil heavy metals [50,51]. Wilck et al. [52] investigated the soil contents of heavy metals in three land use patterns (cultivated land, forest, and grassland) in Slovakia, and found that the concentrations of heavy metals in forest soil was lowest due to complexity of the organic matter. Moreover, the concentrations of Cr, Cu, and nickel (Ni) were highest in cultivated soil, while Cd and Zn were highest in grassland soil [52]. In addition, it was reported that heavy metals under different land use types affect the soil microbial and enzyme mediated soil C/N cycle in karst areas [53]. Our results suggested that the interaction of LUT × SD only had a significant effect on Cu content, and Cd and Cu in different types of land and soil layers had significant statistical differences, indicating that Cd and Cu were the dominant factors controlling soil quality. The contents of Cu and Cd were highest in the top and lower soil layers of *Hylocereus* spp. and *Zanthoxylum planispinum* land while lowest in grassland, implying that agricultural activities including the use of chemical fertilizers, exacerbated the accumulation of heavy metals, which was similar to previous studies [54,55].

There was no significant difference in Cr, Ni, Zn, and Pb contents in 0–20 cm soil layer among the five different types of land use, but the content of these heavy metals was highest in forest in 20–40 cm soil layers, followed by *Zanthoxylum planispinum*, contrary to the reported studies. The studies showed that a large amount of humus in forest has functional groups and chelating quality, which reduces the bioavailability of heavy metals and increases their content. Since woody plants have more roots, the canopy intercepts and absorbs the deposited heavy metals [56]. Therefore, in the process of land use conversion, the environmental effects and the absorption differences of various plants at different spatial and temporal scales produce different results. In general, the contents of heavy metals in grassland, *Hylocereus* spp. and *Zea mays* soil decreased gradually with the increase in soil depth, showing obvious surface enrichment [42], but increased in forest and *Zanthoxylum planispinum* soil. It was reported that Cr content was the highest in deep soil of forest land and grassland, possibly due to the poor migration ability of Cr [57]. The sources and migration characteristics of different heavy metals depend on the type of land use. In a specific ecosystem like karst, the growth of soil microorganisms is affected by vegetation and human activities, and then eventually affects the expression and activity of enzymes [58]. Microorganisms are mostly distributed in the surface soil to decompose the surface litter and root exudates and get more organic matter input [59]. It was reported that soil enzyme activities in temperate grassland and tropical forest, decreased exponentially with depth [60]. The current study found that with the increase in soil depth, the enzyme activity in different types of land gradually decreased. These results were consistent with the findings of Stone et al. [60] and Gelsomino and Azzellino. [61], who believed that increment in the depth resulted in the decreased availability of active substrate and oxygen supply. However, the activities of invertase and urease in the topsoil of grassland and maize were lower than in the deep soil. Grassland as the initial stage of vegetation succession grows rapidly and develops roots. It absorbs a lot of nutrients and improves the decomposition of carbohydrates to stimulate the secretion of sucrase in deep soil [62]. Generally, land use can change soil enzyme activities through plants and microorganisms, or indirectly affect soil enzymes through soil characteristics [63]. In the shallow soil, the activities of Inv, Ure, and Alp were the highest in forest or *Zanthoxylum planispinum* soil, and similar phenomena were also observed in deep soil where activities of Inv, Alp, and Cat were the highest in forest or *Zanthoxylum planispinum* soiland the lowest in *Hylocereus* spp. soil. It was proved that nitrogen fixation of trees again, concurrently, the forest is rich in the litter and the more SOM content, the higher the water holding capacity and effective C, which is more conducive to the activities of microorganisms [64]. A study reported that activities of C and P-related acquisition enzymes increased with the improvement in N utilization, which might explain the increase in Alp and Inv activities in *Zanthoxylum planispinum* soil [64]. In addition, *Hylocereus* spp. was weak in promoting the conversion of invalid P, and effective P nutrition was appropriately supplemented during the cultivation process. However, Cat activity was not significantly different in the surface soil of each land type. Liu et al. [65] said that surface soil SOC and Ca form insoluble substances, resulting in the same inert c pool decomposition of the soil, which is not conducive to the production of oxidase by microorganisms.

In the present study, with the increase in soil depth, the fungal α diversity index (Shannon and Chao 1) showed a downward trend, and the difference was the most significant in the maize field which was consistent with previous studies [66,67]. We also found that except for *Zea mays* soil, the contribution rate of soil depth of the other four land use types to the change of soil fungal community was low. Although some studies showed that there was a little difference between fungal community composition/diversity and soil depth, in the *Zea mays* soil system in karst area, soil depth was the main driving factor of fungal community composition and diversity [68]. The factors affecting fungal diversity are soil management and nutrient level [69]. Our results showed that the Chao1 and Shannon indexes at five different land use types of soil fungal communities were significantly different at 0–20 cm and 20–40 cm soil depths, and the α diversity index was the highest in *Zanthoxylum planispinum* soil and lowest in *Hylocereus* spp. soil. Studies discussed that the ACE, Chao 1 and Shannon indexes of soil fungi in natural shrubs were greater than those in artificial forest and grassland which indicated the importance of vegetation restoration to microbial diversity [70]. The distance of PCoA also showed that land use type mainly had a significant impact on soil fungal community. Different land use types have obvious effects on the structure and diversity of soil fungal community in Karst and non-karst areas which were consistent with our results [71,72]. It was reported that under high-intensity management, soil permeability and nutrient content were increased, which provided a suitable growth environment for soil fungi and improved fungal diversity [73]. However, excessive C and N input caused by high-intensity fertilization could reduce soil microbial diversity, which might explain the lowest fungal diversity in *Hylocereus* spp. and *Zea mays* soil [74].

In different land use patterns, the characteristics of soil fungal communities changed in the karst area. The findings of Cheng et al., were consistent with our results that Ascomycota and Basidiomycota were the dominant phyla in different wetlands and cultivated land, and *Fusarium* was the most dominant genus in the corn field and paddy field in karst [71]. Most of the differences in soil fungal composition are related to soil properties. From grassland and forest to agricultural management land, the abundance of Basidiomycetes decreased, while the abundance of Ascomycota was increased. Ascomycota was mainly involved in the degradation of organic matter and the assimilation of root exudates [75]. Previous studies had investigated that the increase in litter was conducive to the transfer of Basidiomycota to Ascomycota [76]. As an important source of soil microbial nutrients, litter plays an important role in the composition and structure of the microbial community. In addition, the abundance of coccidiota in *Zanthoxylum planispinum* soiland other three types of soil increased significantly. Glomeromycota can form arbuscular mycorrhiza with plants and promote the host to absorb nutrients which indicated that crops need more nutrients [77]. Therefore, the relative abundance of different types of soil fungi was different, indicating the differences in root residues, secretions, and crop management of different plants, which will affect soil physical and chemical properties and then change the species composition and structure of microorganisms.

In the current study, heavy metal content and α and β Diversity of soil fungi were significantly negatively correlated at the soil depths of 0–20 cm in grassland and pepper field, and the results were the same at depths of 20–40 cm soil of *Zanthoxylum planispinum*. Some studies have shown that soil heavy metals can reduce the soil fungal diversity [78]. Mg and K contents had a clear positive effect on soil fungal diversity in forest and *Hylocereus* spp. soil, which may be related to the lack of related elements in these two land types in karst areas. Plenty of studies had confirmed that soil carbon, nitrogen, phosphorus, and pH were the main driving factors to change soil fungal community, it was not clear whether soil mineral nutrients would affect the soil fungal community structure [79]. The results of RDA analysis demonstrated that the relationship between fungal phylum communities and soil mineral content in different land use types was different. In different soil depths of *Zea mays* and *Hylocereus* spp. soil, the dominant fungi were significantly correlated with K and Na. Pan et al., found that total K was positively correlated with Ascomycota and Basidiomycota, and soil K could provide nutrients for soil microorganisms [80]. There was a significant positive correlation between Basidiomycetes and content of Mg, Ca and Fe at the depths of 20–40 cm soil in grassland, which was consistent with previous studies on the relationship between soil fungal community composition and soil Fe content [81]. The correlation between soil mineral nutrients and fungal community structure in karst areas was different in five land types, which may be caused by different soil environments in the karst region.

Fungi are considered to be the main producers of soil enzymes. It was reported that the relationship between soil fungal community diversity and soil enzyme activity [82,83]. In our study, soil fungi with different use types and depths were correlated with different kinds of enzymes. Inv and Alp mainly affect the fungal community in forest and *Zanthoxylum planispinum* soil. Studies have shown that alkaline phosphatase was mainly related to the fungal composition of grassland topsoil and mature forest subsoil [83]. Rotation and fertilization will increase the enzymes involved in organic carbon, nitrogen mineralization and decomposition. Our results showed that Ure and Inv activity mainly affected the composition and diversity of fungi in *Zanthoxylum planispinum* and *Zea mays* soil [84]. Therefore, farmland management can regulate the effect of soil fungal communities on soil enzyme activity. Similarly, we also found that Inv was positively correlated with Glomeromycota and Kickxellomycota. Therefore, these two fungi might be the main fungal species for the decomposition of soil organic matter in forest and grassland. However, some studies have shown that the relativity between soil enzyme activity and abiotic factors is greater than that with fungal community, so it is necessary to further study the relationship between soil enzyme and abiotic factors such as soil physical and chemical properties [85].

## 5. Conclusions

Rational land use is conducive to improving soil nutrients and soil fungal communities in karst areas. In this study, land use type and soil depth significantly affected soil mineral elements contents, soil enzyme activity, and fungal community. Specifically, land use types had significant effects on the contents of soil K, Mg, Fe, Cu and Cr; however, soil depth had no significant effect on soil mineral elements contents. Both land use type and depth significantly affected the invertase, urease, alkaline phosphatase, and catalase activity. In addition, Shannon and Chao1 index of soil fungal community was affected by different land use types and depths. Ascomycota, Basidiomycota, and Mortierellomycota were the dominant phyla at 0–20 cm and 20–40 cm soil depths on five different land types. However, soil depth had no significant effect on the soil fungal structure. This might be because small-scale environmental variation in the karst areas was not the dominant factor in changing the fungal community structure. Soil mineral elements content, enzyme activity, and soil fungal community in the karst area were strongly affected by land use type and soil depth. Our findings provided a theoretical basis for the rational use of limited land in Karst’s fragile ecological environment. More importantly, in recent years, the local government has encouraged farmers to grow different cash crops, which has led to changes in land types and management. The current research can help to guide the selection of appropriate land use types for crop planting and nutrient management in karst areas based on soil fungal community structure.

## Figures and Tables

**Figure 1 ijerph-19-03120-f001:**
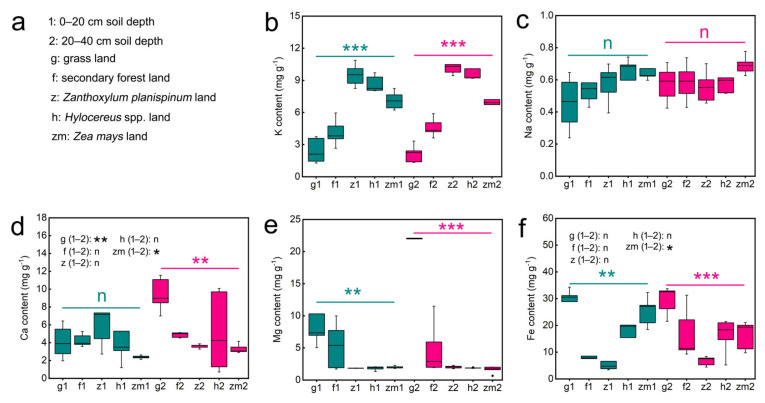
Effect of the different land use types on content of soil K, Na, Ca, Mg Fe in 0–20 cm and 20–40 cm soil depth. (**a**) the description of abbreviations, (**b**) K content, (**c**) Na content, (**d**) Ca content, (**e**) Mg content, (**f**) Fe content. ** and *** above green line and purple line indicate there are significant difference at *p* < 0.05, *p* < 0.01 and *p* < 0.001 level between the five different land use types in 0–20 cm and 20–40 cm soil depths, respectively; n indicates that there is no statistically significantly different among the five different land use types. * and ** after g/f/z/h/zm (1–2) indicate significant difference at *p* < 0.05 and *p* < 0.01 level between 0–20 cm and 20–40 cm depths in the five different land use types, respectively, n indicates not statistically significantly different between 0–20 cm and 20–40 cm soil depths in the five different land use types, respectively.

**Figure 2 ijerph-19-03120-f002:**
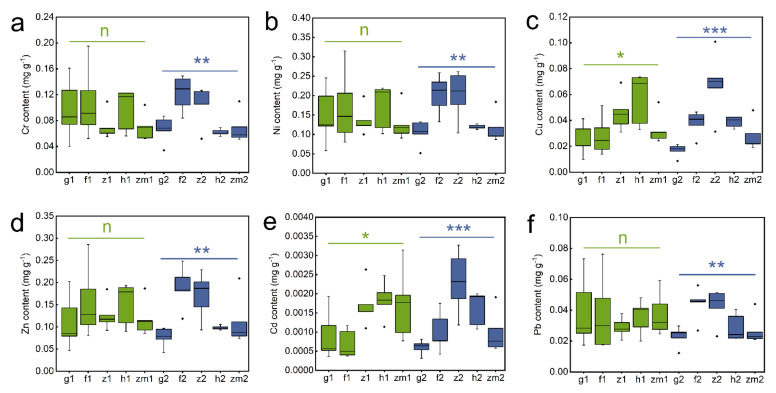
Effect of the different land use types on contents of Cr, Ni, Cu, Zn, Cd, Pb in 0–20 cm and 20–40 cm soil depth. (**a**) Cr content, (**b**) Ni content, (**c**) Cu content, (**d**) Zn content, (**e**) Cd content, (**f**) Pb content. *, ** and *** above green line and blue line indicate there are significant difference at *p* < 0.05, *p* < 0.01 and *p* < 0.001 level between the five different land use types in 0–20 cm and 20–40 cm soil depths, respectively; n indicates that there is no statistically significantly different among the five different land use types.

**Figure 3 ijerph-19-03120-f003:**
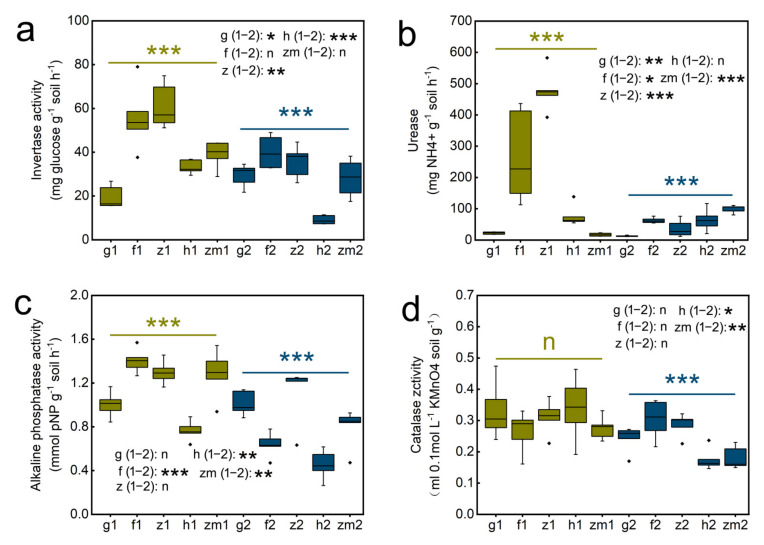
Effect of the different land use types on soil enzyme activity in 0–20 cm and 20–40 cm soil depth. (**a**) invertase (Inv) activity (**b**) urease (Ure) activity, (**c**) alkaline phosphatase (Alp) activity, (**d**) catalase (Cat) activity. *** above yellow line and blue line indicate there were significant difference at *p* < 0.001 level between the five different land use types in 0–20 cm and 20–40 cm soil depths, respectively; n indicates that there is no statistically significantly different among the five different land use types. *, ** and *** after g/f/z/h/zm (1–2) indicate significant difference at *p* < 0.05, *p* < 0.01 and *p* < 0.001 level between 0–20 cm and 20–40 cm depths at the five different land use types, respectively, n indicates not statistically significantly different between 0–20 cm and 20–40 cm soil depths at the five different land use types, respectively. The abbreviations are described in Figure 1a.

**Figure 4 ijerph-19-03120-f004:**
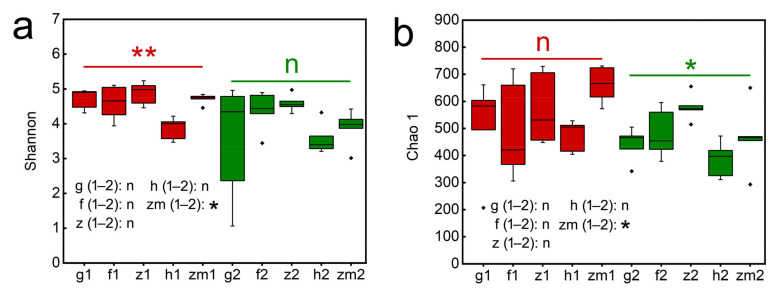
Effect of the different land use types on fungal alpha diversity, (**a**) Chao1, (**b**) Shanon in 0–20 cm and 20–40 cm soil depth. * and ** above red line and green line indicate there are significant difference at *p* < 0.05 and *p* < 0.01 level between the five different land use types in 0–20 cm and 20–40 cm soil depths, respectively; n indicates that there is no statistically significantly different among the five different land use types. * after g/f/z/h/zm (1–2) indicates significant difference at *p* < 0.05 level between 0–20 cm and 20–40 cm depths at the five different land use types, respectively, n indicates not statistically significantly different between 0–20 cm and 20–40 cm soil depths at the five different land use types, respectively. The abbreviations are described in Figure 1a.

**Figure 5 ijerph-19-03120-f005:**
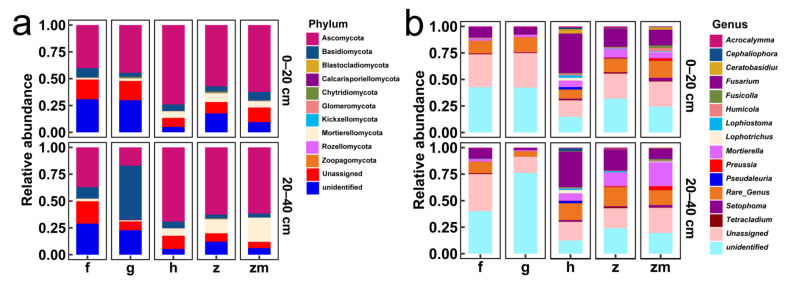
The relative abundance of major fungal phyla in all soil samples ((**a**,**b**) represent phylum level and genus level %, respectively). The abbreviations are described in Figure 1a.

**Figure 6 ijerph-19-03120-f006:**
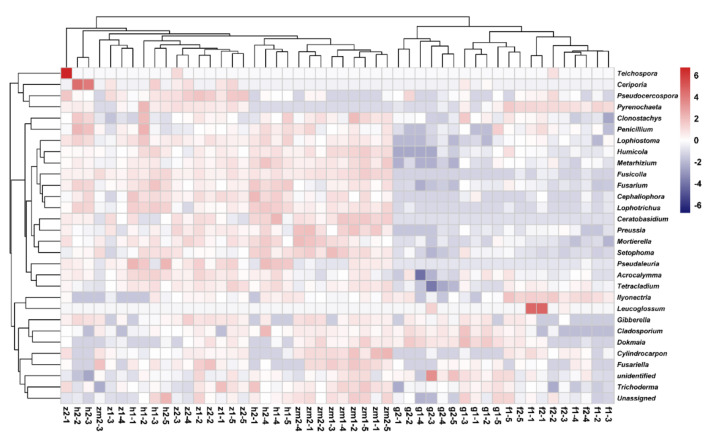
Heat maps of the 30 most abundant fungal genera in different land use types at 0–20 cm and 20–40 cm soil depths, the absolute abundance of fungi is expressed by color intensity. The abbreviations are described in Figure 1a.

**Figure 7 ijerph-19-03120-f007:**
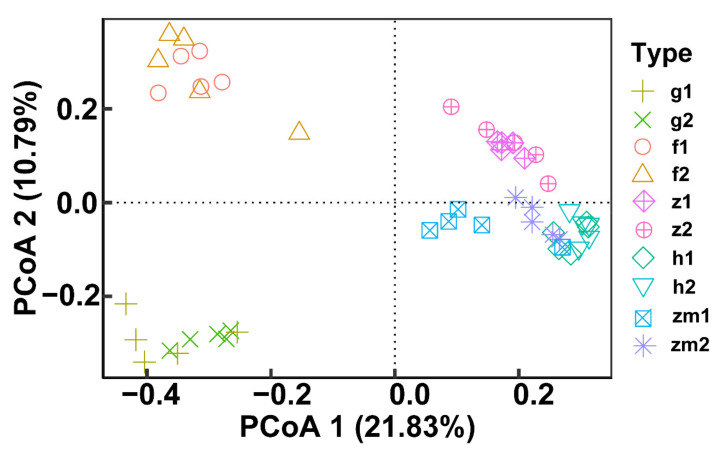
Effect of the different land use types on fungal beta diversity at 0–20 cm and 20–40 cm soil depths. Principal coordinate analysis (PCoA) based on Bray-Curtis of all soil fungal communities. The abbreviations are described in Figure 1a.

**Figure 8 ijerph-19-03120-f008:**
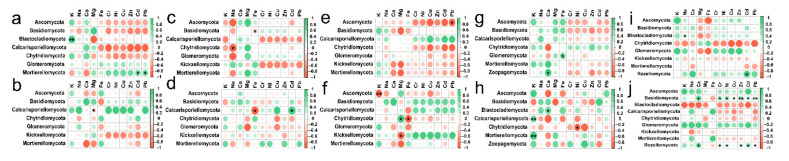
Spearman’s Rank correlation coefficients between soil mineral elements content and soil fungal abundance at (**a**) 0–20 cm in grassland, (**b**) 20–40 cm in grassland, (**c**) 0–20 cm in forest, (**d**) 20–40 cm in forest, (**e**) 0–20 cm in *Zanthoxylum planispinum* soil, (**f**) 20–40 cm in *Zanthoxylum planispinum* soil, (**g**) 0–20 cm in *Hylocereus* spp. soil, (**h**) 20–40 cm in *Hylocereus* spp. soil, (**i**) 0–20 cm in *Zea mays* soil and (**j**) 20–40 cm in *Zea mays* soil. * and ** indicate obvious difference at *p* < 0.05 and *p* < 0.01 between soil mineral elements content and the abundance of soil fungal phyla.

**Figure 9 ijerph-19-03120-f009:**
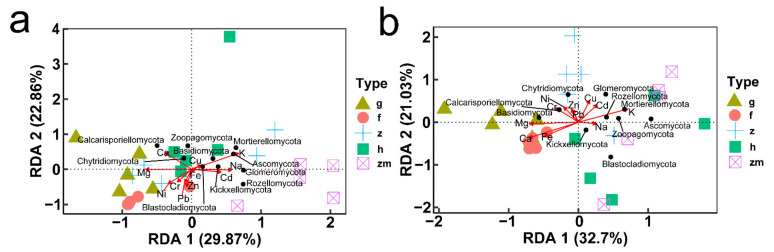
Redundancy analysis (RDA) of relationship between soil mineral elements content (**red arrows**) and the relative abundance of soil microbial phyla (**black points**) at (**a**) 0–20 cm soil depth and (**b**) 20–40 cm soil depth, respectively. The abbreviations are described in Figure 1a.

**Figure 10 ijerph-19-03120-f010:**
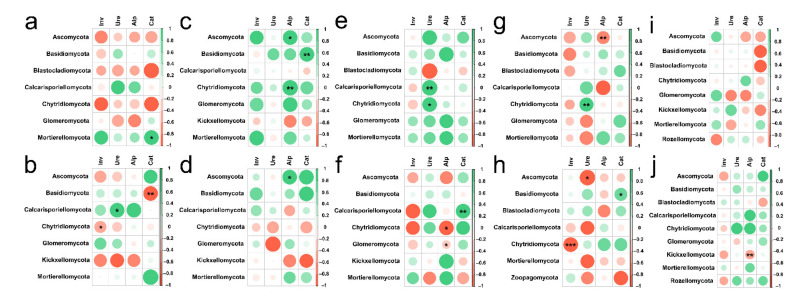
Spearman’s Rank correlation coefficients between soil enzyme activity and soil fungal abundance at (**a**) 0–20 cm in grassland, (**b**) 20–40 cm in grassland, (**c**) 0–20 cm in forest, (**d**) 20–40 cm in forest, (**e**) 0–20 cm in *Zanthoxylum planispinum* soil, (**f**) 20–40 cm in *Zanthoxylum planispinum* soil, (**g**) 0–20 cm in *Hylocereus* spp. soil, (**h**) 20–40 cm in *Hylocereus* spp. soil, (**i**) 0–20 cm in *Zea mays* soil and (**j**) 20–40 cm in *Zea mays* soil. *,** and *** indicate obvious difference at *p* < 0.05, *p* < 0.01 and *p* < 0.001 between soil enzyme activity and the abundance of soil fungal phyla.

**Figure 11 ijerph-19-03120-f011:**
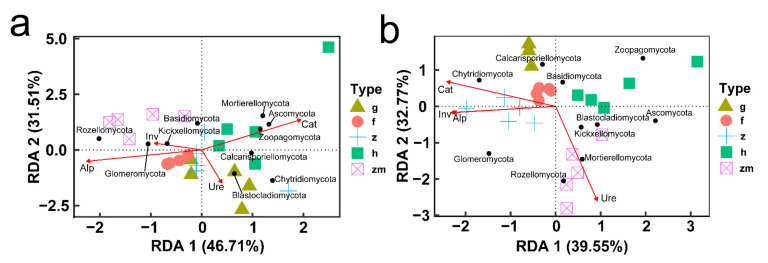
Redundancy analysis (RDA) of relationship between soil enzyme activity (**red arrows**) and the richness of soil microbial phyla (**black points**) at (**a**) 0–20 cm soil depth and (**b**) 20–40 cm soil depth, respectively. The abbreviations are described in Figure 1a.

**Figure 12 ijerph-19-03120-f012:**
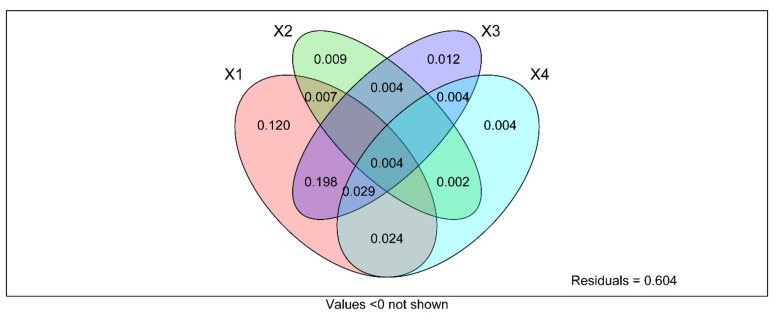
Variance distribution analysis (VPA) to determine the relative contributions of land use type, soil depth, soil metal element content and soil enzyme activity to soil fungal community structure and diversity. X1 = LUT, X2 = SD, X3 = soil metal element content, X4 = soil enzyme activity. Value represents the significance level when *p* < 0.0 not shown.

## Data Availability

Not applicable.

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
