# Peer review of "Effects of Different Land Use Types and Soil Depths on Soil Mineral Elements, Soil Enzyme Activity, and Fungal Community in Karst Area of Southwest China"

_ijerph, 2022, doi:10.3390/ijerph19053120_

Round 1

Reviewer 1 Report

Attached are some sentences and parts of the text that you need to correct. My opinion is that the authors can use the results but they must change the way the text is presented and seek professional help with the English language

Author Response

Dear Reviewer,

We have revised the manuscript “Effects of different land use types and soil depths on soil metal content, soil enzyme activity and fungal community in karst area of Southwest China” (ijerph-1602648) according to the reviewer’s comments. In our point-by-point responses attached below, reviewers’ comments are in black font and our responses are in blue font.

First of all, thank you very much for your great help and good suggestions regarding our manuscript. In our point-by-point responses below, and the reviewer’s comments are in black font and our responses are in blue font.

Comments for the Author:

Reviewer #1: Attached are some sentences and parts of the text that you need to correct. My opinion is that the authors can use the results but they must change the way the text is presented and seek professional help with the English language

First of all, thank you very much for your great help, patient guidance and good suggestions regarding our manuscript. We have carefully examined our manuscript and have improved the grammar and spelling, and the writing quality has been improved. We hope that the quality of the writing of this manuscript is now satisfactory.

Reviewer #1: I am not authoritative to evaluate the language, but it is obvious that this manuscript needs language correction. Everything marked in yellow, below, is not well defined, not clear, there are errors in the language. Please correct.

Thank you very much for your attention and affirmation of the research content of the manuscript and your valuable suggestions for my contributions. We have carefully examined our manuscript and have improved the grammar and spelling, and the writing quality has been improved. We hope that the quality of the writing of this manuscript is now satisfactory.

Reviewer #1: row 24 - When you write Zanthoxylum planispinum land, Hylocereus spp. land and Zea mays land it means that it is the name of the type of land, not the land on which the plant is grown. Please change that.

Thank you very much for your patient and good suggestions. We have carefully corrected this unprofessional term.

Reviewer #1: row 42 - Due to the unique geographical conditions, the southwest karst region centered on Guizhou Province in China is the world's largest continuous coverage area, about 5.1million km2 , the most complete type of development, the most typical karst ecosystem. Modify the yellow part.

Thank you very much for your patient and good suggestions. We have carefully corrected the parts marked in yellow.

Reviewer #1: row 72 - In addition, alkaline phosphatase, it plays an important role in organic phosphorus (P) mineralization and plant P nutrition, especially in calcareous soil with limited P. Modify the yellow part.

Thank you very much for your patient and good suggestions. We have carefully corrected the parts marked in yellow.

Reviewer #1: row 74 - And microbial activity is a key indicator to detect soil quality and control soil degradation which fully show that soil microorganism is still an important research direction, especially in karst areas. Modify the yellow part.

Thank you very much for your patient and good suggestions. We have carefully corrected the parts marked in yellow.

Reviewer #1:row 82 - Heavy metals are widespread in the surface environment which are persistent, stable and not easy to degrade. Modify the yellow part.

Thank you very much for your patient and good suggestions. We have carefully corrected the parts marked in yellow.

Reviewer #1: row 83 - In recent years, due to the unreasonable development of mineral resources and improper disposal of hazardous wastes, coupled with the extreme vulnerability of groundwater system, heavy metals in karst areas have been seriously diffused, which has posed a threat to the biological community. Modify the yellow part.

Thank you very much for your patient and good suggestions. We have carefully corrected the parts marked in yellow.

Reviewer #1: row 94 - Based on the above tissues, the contents of soil metals, soil enzyme activities and fungal communities in five land use types and two soil depths in Guizhou Province of China were investigated, in order to provide theoretical basis for soil management and ecological restoration in karst areas. The term “soil metals” is not acceptable. Modify the yellow part.

Thank you very much for your patient and good suggestions. We have carefully corrected the parts marked in yellow.

Reviewer #1: row 104 - the annual mean rainfall is about 1200 mm yr-1. Modify the yellow part.

Thank you very much for your patient and good suggestions. We have carefully corrected the parts marked in yellow.

Reviewer #1: row 120 - the content of soil metal elements. The term “soil metal elements” is not acceptable.

Thank you very much for your patient and good suggestions. We have carefully changed the inappropriate term of “soil metal elements” to “soil mineral elements content”.

Reviewer #1: row 124 - flame atomization method. Modify the yellow part.

Thank you very much for your patient and good suggestions. We have carefully corrected the parts marked in yellow.

Reviewer #1: row 127 - atomic absorption spectrometric. Modify the yellow part.

Thank you very much for your patient and good suggestions. We have carefully corrected the parts marked in yellow.

Reviewer #1: row 128 - (detection limit is 0- 2000 μg/L). Detection limit cannot be “0”. You cannot show the detection limit for these metals in this way

Thank you very much for your patient and good suggestions. We have removed this sentence.

Reviewer #1: row 157 - And statistical significance was defined as P=0.05 confidence level, and the mean was evaluated by standard error. Modify the yellow part.

Thank you very much for your patient and good suggestions. We have carefully corrected the parts marked in yellow.

Reviewer #1: row 159 - The Chao1, Shannon was calculated with by R (version 3.2.2).

Thank you very much for your patient and good suggestions. We have carefully corrected the parts marked in yellow.

Reviewer #1: row 173 - Zea mays land soils. Modify the yellow part.

Reviewer #1: row 180, 182, 184 - Zanthoxylum planispinum land soils, Zea mays land soil. When you write Zanthoxylum planispinum land, Hylocereus spp. land and Zea mays land it means that it is the name of the type of land, not the land on which the plant is grown. Please change that in whole text.

I'm sorry for my carelessness and unclear expression. We have carefully corrected the parts.

Reviewer #1: row 199 - Subsequent studies have shown that under the five land use types, there was no significant difference in heavy metal content between different layers. Modify the yellow part.

Thank you very much for your patient and good suggestions. We have carefully corrected the parts marked in yellow.

Reviewer #1: row 262 - but the interaction of LUT and SD had no effect on the Shannon. Modify the yellow part.

Thank you very much for your patient and good suggestions. We have carefully corrected the parts marked in yellow.

Reviewer #1: row 265 - Our results also showed that of Shannon and Chao 1 were affected by…Modify the yellow part.

Thank you very much for your patient and good suggestions. We have carefully corrected the parts marked in yellow.

Reviewer #1: row 272 - The same result was shown in the Chao 1, Chan 1 under 0-20 cm soil depth in Zanthoxylum planispinum land was the highest (579.20 ± 22.39) and Hylocereus spp. land was the lowest (385.00 ± 29.85) (Figure 4b). What is chan 1? And totally confusing.

I'm sorry for my carelessness and unclear expression. “Chan 1” should be changed to “Chao 1”, I have modified it in row 272. And (579.20 ± 22.39) represent the value and standard error of Chao1 in 0-20 cm depth soil in Zanthoxylum planispinum, respectively. Similarly, (385.00 ± 29.85) represent the value and standard error of Chao1 in 0-20 cm depth soil in Hylocereus spp., respectively.

Reviewer #1: row 309 - The five land types were significantly dispersed at the depth of 0-20 cm and 20-40 cm, respectively, but except for Zea mays land, compared with 0-20 cm, points of 20-40 cm can be clustered accurately, indicating the similarity of fungal community composition in upper and lower soil layers. A very long sentence, I didn't understand what you meant.

I'm sorry for my lengthy and unclear expression. We have corrected it to “The first principal component explained 21.83% of the total variance, and the second principal component explained 10.79% of the variance. The points of five different land types were significantly dispersed, indicating that land use type caused the significant change of soil fungal community structure. However, except for Zea mays, the points at 0-20 cm depths soil and 20-40 cm depths soil were very concentrated, indicating the similarity of fungal community composition at the two soil depths, and the effect of soil depth on soil fungal structure was not significant”.

Reviewer #1: row 327 - only one fungal genus was significantly richer than the upper soil. Modify the yellow part.

Thank you very much for your patient and good suggestions. We have carefully corrected the parts marked in yellow.

Reviewer #1: row 329 - the richness of Glomeromycota and Glomus were significantly higher than that in the lower soil (Figure S1b). In Hylocereus spp. land, the richness of Tecladium in 0-20 cm soil was significantly higher than that in 20-40 cm soil (Figure S1c). In Zea mays land, the richness of Humicola in 0-20 cm soil layer was significantly higher than that in 20-40 cm soil layer, while the richness of Gibberella in 20-40cm soil layer was significantly higher than that in 0-20 cm soil layer (Figure S1d). We also found that there were no fungal groups with significant differences in richness in different soil depths of forest. “Richness” is not acceptable term.

Thank you very much for your patient and good suggestions. We have carefully changed the inappropriate term of “Richness” to “relative abundance”.

Reviewer #1: row 337 - Relevance of soil fungal communities and diversity with content of soil mineral elements. Relevance is not acceptable term.

Thank you very much for your patient and good suggestions. We have carefully changed the inappropriate term of “Relevance” to “Correlation”.

Reviewer #1: row 338 - there were significant relevance between soil fungal diversity and soil mineral contents of different land use types. Modify the yellow part.

Thank you very much for your patient and good suggestions. We have carefully corrected the parts marked in yellow.

Reviewer #1: row 340 - under 0-20 cm soil depth. Modify the yellow part.

Thank you very much for your patient and good suggestions. We have carefully corrected the parts marked in yellow.

Reviewer #1: row 353 - Under 0-20 cm soil depth. Modify the yellow part.

Thank you very much for your patient and good suggestions. We have carefully corrected the parts marked in yellow.

Reviewer #1: row 359 - RDA also showed that in upper soil depth. “upper” is not acceptable term.

Thank you very much for your patient and good suggestions. We have carefully changed the inappropriate term of “upper” to “topsoil layer”.

Reviewer #1: row 364 - Under 20-40 cm soil depth. Please check everywhere in the text where you use the term “under”.

Thank you very much for your patient and good suggestions. We have carefully corrected all uses of this inappropriate term in this manuscript.

Reviewer #1: row 384 - Relevance of soil fungal communities and diversity with soil enzyme activity. Modify the yellow part.

Thank you very much for your patient and good suggestions. We have carefully corrected the parts marked in yellow.

Reviewer #1: row 440 - more than 6 × 108 hm2. Modify the yellow part.

Thank you very much for your patient and good suggestions. We have carefully corrected the parts marked in yellow.

Reviewer #1: row 441 - The land resources in karst area are limited and the ecological environment is fragile. In order to establish a sustainable land use system, the present research studied the differences of soil mineral nutrients and heavy metal contents, soil enzyme activities and soil fungal communities under five different land use types and two soil depths in Karst area, Southwest China, the phylum composition of soil fungal community was also analyzed. This sentence should not be part of the Discussion.

Thank you very much for your patient and good suggestions. We have removed the parts in “Discussion”.

Reviewer #1: row 447 - and soil mineral elements is affected by soil properties and land. Modify the yellow part.

Thank you very much for your patient and good suggestions. We have carefully corrected the parts marked in yellow.

Reviewer #1: row 448 - A study showed that grassland reclamation in karst area reduced soil trace elements (Cu, Fe, Mo, B) and presented enrichment effect in surface soil. Modify the yellow part.

Thank you very much for your patient and good suggestions. We have carefully corrected the parts marked in yellow.

Reviewer #1: row 457 - K content in cultivated land soil. Change everywhere in the text where this expression appears.

I'm sorry for my carelessness and unclear expression. We have carefully corrected this unprofessional term

Reviewer #1: row 466 - The input of litter is related to the abundance of trees, which not only contributes to the low variability of C and N concentration on a small scale, but also makes the trees that absorb Ca from deep soil maintain high content of available Ca, thus contributing to local reduction of Ca. Confusing sentence.

I'm sorry for my carelessness and unclear expression. We have corrected it to “The input of litter is related to the richness of the forest. The plant tissue falling from tulip poplar constituted a large amount of organic matter input into the soil, which contributes to the low variability of C and N concentration on a small scale, while the Ca accumulated in plant tissues enables the trees to absorb it from deep soil layers to maintain the effective calcium concentration in topsoil layers, which eventually cause the reduction of local soil Ca variability”.

Reviewer #1: row 488 - Land use types correspond to the relevant changes of management, soil texture, vegetation type, microbial activity, these changes have an important impact on the migration, transformation and enrichment of soil heavy metals, and even lead to soil heavy metal pollution. Too long sentence.

I'm sorry for my lengthy and unclear expression. We have corrected it to “Land use types correspond to the changes in soil management, vegetation types and microbial activity. These changes have an important impact on the migration, transformation and enrichment of soil heavy metals”.

Reviewer #1: row 624 - However, some studies have shown that the relativity between soil enzyme activity and abiotic factors is greater than that with fungal community, so it is necessary to further study the relationship between soil enzyme and properties of soil physical and chemical. Modify the yellow part.

Thank you very much for your patient and good suggestions. We have carefully corrected the parts marked in yellow.

English language and style

Reviewer #1: Extensive editing of English language and style required

Firstly, thank you for your valuable suggestions. Our manuscript has been critically read and revised by a native English speaking coauthor. We hope that the writing of this manuscript is now satisfactory.

Reviewer #2: Moderate English changes required

Firstly, thank you for your valuable suggestions. Our manuscript has been critically read and revised by a native English speaking coauthor. We hope that the writing of this manuscript is now satisfactory.

Reviewer's Responses to Questions

  1. Does the introduction provide sufficient background and include all relevant references?

Reviewer #1: Must be improved

Thank you very much for your good suggestion. We have carefully re-evaluated our manuscript and revised the content including the “Introduction” to be more able to provide sufficient context about our study to make the manuscript clearer for readers.

  1. Is the research design appropriate?

Reviewer #1: Yes

Thank you for your positive assessment.

  1. Are the methods adequately described?

Reviewer #1: Can be improved

Thank you very much for your good suggestion. We have carefully re-evaluated our manuscript and revised the content, including the “Materials and Methods”, to be more informative and specific to provide clarity to the reader.

  1. Are the results clearly presented?

Reviewer #1: Must be improved

Thank you very much for your good suggestion. We have carefully re-evaluated our manuscript and revised the content, including the “Results”, to be more informative and specific to provide clarity to the reader.

  1. Are the conclusions supported by the results?

Reviewer #1: Can be improved

Thank you very much for your good suggestion. We have carefully re-evaluated our manuscript and revised the experimental design section to make the descriptions in this section more intuitive and specific in order to provide clear information to the reader.

Reviewer 2 Report

I have a few significant queries regarding the experimental design.

"Each land use type was repeated 5 times, and a total of 25 land use
types were obtained.
" - So only 5 samples within each land use type, or 5 different field plots per land use type? That is not sufficient replication within sites. Each individual sample appears to be 1 cubic metre of soil, which is an extraordinary volume, with no mention of sub-sampling? For the soil tests described, 1,000s of sub-samples could have been taken.

"the soil samples from 5 different types of land use (4 m × 4 m) were collected, including: Zanthoxylum planispinum land, Hylocereus spp. land, Zea mays land, grassland (the main species are Themeda japonica) and secondary forest land (the main species are Liquidambar formosana)" - This also doesn't make sense. Is each vegetation type only 16 sq metres? How did you define and distinguish each land use?

A map showing where your field work was conducted would be highly useful to see how far away each site is and to better understand the sampling design applied. Please include this map, or detail specifically where you conducted your field work.

Regarding novelty, I am not sure what this paper is saying. Different land types have different soil chemistry and communities. That seems intuitive? Especially given you're comparing corn fields with forests - The soils would have been extensively modified in farmlands. I would consider really reframing the significance of this work, and why the design of this study is adding new and important knowledge for land management in Karst systems.

In the conclusions, there's mention of improving soil nutrients for better outcomes in Karst systems. I recommend this paper for reading and perhaps citing as it has similar themes regarding nutrients, and overlaps with a few elements mentioned in this study - https://www.mdpi.com/1999-4907/11/8/797

The English requires more proof-reading throughout. It's not too bad, but it doesn't flow well in a lot of sections due to awkward phrasing.

Author Response

Dear Reviewer,

We have revised the manuscript “Effects of different land use types and soil depths on soil metal content, soil enzyme activity and fungal community in karst area of Southwest China” (ijerph-1602648) according to the reviewer’s comments. In our point-by-point responses attached below, reviewers’ comments are in black font and our responses are in blue font.

First of all, thank you very much for your great help and good suggestions regarding our manuscript. In our point-by-point responses below, and the reviewer’s comments are in black font and our responses are in blue font.

Reviewer #2: I have a few significant queries regarding the experimental design. “Each land use type was repeated 5 times, and a total of 25 land use types were obtained” - So only 5 samples within each land use type, or 5 different field plots per land use type? That is not sufficient replication within sites. Each individual sample appears to be 1 cubic metre of soil, which is an extraordinary volume, with no mention of sub-sampling? For the soil tests described, 1,000s of sub-samples could have been taken.

I'm sorry for my unclear expression. We have carefully examined our manuscript and have restructured and revised this part to make the contents clearer for the readers. The description of our sampling method is: “There were 5 sampling points for each land use type, the area of each sampling point is 4 m × 4 m, and the distance between each sampling point was 5 m. After removing 1-2 cm of topsoil, soil samples were collected at depths of 0-20 cm and 20-40 cm, respectively. Each soil sample was mixed from four subplots (1 m x 1 m range) at 0-20 cm and 20-40 cm soil depths, respectively, as a replicate, and collected in dry, clean, sterile polyethylene bags…”.

Reviewer #2: “the soil samples from 5 different types of land use (4 m × 4 m) were collected, including: Zanthoxylum planispinum land, Hylocereus spp. land, Zea mays land, grassland (the main species are Themeda japonica) and secondary forest land (the main species are Liquidambar formosana) - This also doesn't make sense. Is each vegetation type only 16 sq metres? How did you define and distinguish each land use?
I'm sorry for my unclear expression. The area of the sampling point selected for each land use type is 4 m × 4 m, which is the maximum area projected to the ground according to the plants growing on each land use type. And we define and distinguish these five different land use types according to the main plant species on each land use type: grassland (the main species are Themeda japonica), secondary forest (the main species are Liquidambar formosana); pepper field (the main cultured species are Zanthoxylum planispinum), dragon fruit field (the main cultured species is Hylocereus spp.) and maize field (the main cultured species are Zea mays).

Reviewer #2: A map showing where your field work was conducted would be highly useful to see how far away each site is and to better understand the sampling design applied. Please include this map, or detail specifically where you conducted your field work.

Thank you very much for your patient and good suggestions. We have added the elevation, latitude and longitude of the sampling sites of Huajiang town in Guanling Buyi and Miao Autonomous county, Anshun city, Guizhou province, southwest China with an area of 294.9 km2 (Altitude 1439m, 105°34’E, 25°43’N) and the specific location information for each land use type has also been supplemented.

Reviewer #2: Regarding novelty, I am not sure what this paper is saying. Different land types have different soil chemistry and communities. That seems intuitive? Especially given you're comparing corn fields with forests - The soils would have been extensively modified in farmlands. I would consider really reframing the significance of this work, and why the design of this study is adding new and important knowledge for land management in Karst systems.
Thank you very much for your patient and good suggestions. As one of the most special ecosystems in the world, the karst area has a fragile environment and limited available land area. Therefore, the rational use of land in karst areas is particularly important. For the first time, we investigated the effects of land use type and soil depth on soil mineral elements content, soil enzyme activity and soil fungal community in a karst area. Our results confirmed that land use type significantly affected soil K, Mg, Fe, Cu, and Cr contents, and land use type × soil depth significantly affected the activities of invertase, urease, alkaline phosphatase, and catalase. In addition, the Shannon and Chao1 index of soil fungal communities were affected by different land use types and soil depths. Land use also caused significant changes in the soil fungal structure. We also explored the relationship between soil mineral nutrients and enzyme activities to varying degrees with different soil fungal communities. In conclusion, land use type and soil depth significantly affected soil mineral element nutrients, soil enzyme activities and soil fungal community structure, and there were obvious correlations. Our findings highlight the responses of soil fungal communities to soil depths and land use types in the karst areas, aiming to find the key factors driving fungal community changes in different land use types. In recent years, the local government has encouraged farmers to grow different cash crops, which has led to about changes in land types and management. Our research can help to guide the selection of appropriate land use types for crop planting and nutrient management in the karst areas based on soil fungal community structure.

Reviewer #2: In the conclusions, there's mention of improving soil nutrients for better outcomes in Karst systems. I recommend this paper for reading and perhaps citing as it has similar themes regarding nutrients, and overlaps with a few elements mentioned in this study - https://www.mdpi.com/1999-4907/11/8/797

Thank you very much for your patient and good suggestions. We have read this article carefully and the research methods and conclusions in it are helpful for our manuscript.

Reviewer #2: The English requires more proof-reading throughout. It's not too bad, but it doesn't flow well in a lot of sections due to awkward phrasing.

Thank you very much for your great help, patient guidance and good suggestions. Our manuscript has been critically read and revised by a native English speaking coauthor, and the writing quality has been improved. We hope that the quality of the writing of this manuscript is now satisfactory.

English language and style

Reviewer #1: Extensive editing of English language and style required

Firstly, thank you for your valuable suggestions. Our manuscript has been critically read and revised by a native English speaking coauthor. We hope that the writing of this manuscript is now satisfactory.

Reviewer #2: Moderate English changes required

Firstly, thank you for your valuable suggestions. Our manuscript has been critically read and revised by a native English speaking coauthor. We hope that the writing of this manuscript is now satisfactory.

Reviewer's Responses to Questions

  1. Does the introduction provide sufficient background and include all relevant references?

Reviewer #2: Yes

Thank you for your positive assessment.

  1. Is the research design appropriate?

Reviewer #2: Must be improved

Thank you very much for your good suggestion. We have carefully re-evaluated our manuscript and improved the experimental design, to be more informative and specific to provide clarity to the reader.

  1. Are the methods adequately described?

Reviewer #2: Must be improved

Thank you very much for your good suggestion. We have carefully re-evaluated our manuscript and revised the content, including the “Materials and Methods”, to be more informative and specific to provide clarity to the reader.

  1. Are the results clearly presented?

Reviewer #2: Can be improved

Thank you very much for your good suggestion. We have carefully re-evaluated our manuscript and revised the content, including the “Results”, to be more informative and specific to provide clarity to the reader.

  1. Are the conclusions supported by the results?

Reviewer #2: Yes

Thank you for your positive assessment.

Round 2

Reviewer 1 Report

The authors accepted the suggestions and corrected parts of the text as suggested.

This manuscript is a resubmission of an earlier submission. The following is a list of the peer review reports and author responses from that submission.